# Energetic demands regulate sleep-wake rhythm circuit development

**Amy R Poe[1], Lucy Zhu[1], Si Hao Tang[1], Ella Valencia[1], Matthew S Kayser[1,2,3]***

[1]Department of Psychiatry, Perelman School of Medicine, University of Pennsylvania, Philadelphia, United States; [2]Chronobiology and Sleep Institute, Perelman School of Medicine, University of Pennsylvania, Philadelphia, United States; [3]Department of Neuroscience, Perelman School of Medicine, University of Pennsylvania, Philadelphia, United States

**Abstract** Sleep and feeding patterns lack strong daily rhythms during early life. As diurnal animals mature, feeding is consolidated to the day and sleep to the night. In *Drosophila*, circadian sleep patterns are initiated with formation of a circuit connecting the central clock to arousal output neurons; emergence of circadian sleep also enables long-term memory (LTM). However, the cues that trigger the development of this clock-arousal circuit are unknown. Here, we identify a role for nutritional status in driving sleep-wake rhythm development in *Drosophila* larvae. We find that in the 2nd instar larval period (L2), sleep and feeding are spread across the day; these behaviors become organized into daily patterns by the 3rd instar larval stage (L3). Forcing mature (L3) animals to adopt immature (L2) feeding strategies disrupts sleep-wake rhythms and the ability to exhibit LTM. In addition, the development of the clock (DN1a)-arousal (Dh44) circuit itself is influenced by the larval nutritional environment. Finally, we demonstrate that larval arousal Dh44 neurons act through glucose metabolic genes to drive onset of daily sleep-wake rhythms. Together, our data suggest that changes to energetic demands in developing organisms trigger the formation of sleep-circadian circuits and behaviors.

**\*For correspondence:**
kayser@pennmedicine.upenn.edu

## eLife assessment

This study presents **valuable** findings as it shows that sleep rhythm formation and memory capabilities depend on a balanced and rich diet in fly larvae. The evidence supporting the claims of the authors is **convincing** with rigorous behavioral assays and state-of-the-art genetic manipulations. The work will be of interest to researchers working on sleep and memory.

## Introduction

The development of behavioral rhythms such as sleep-wake patterns is critical for brain development (*Vallone et al., 2007*). Indeed, early life circadian disruptions in rodents negatively impacts adult behaviors, neuronal morphology, and circadian physiology (*Varcoe et al., 2016*; *Mendez et al., 2016*; *Ameen et al., 2022*). Likewise, in humans, disruptions in sleep and rhythms during development are a common co-morbidity in neurodevelopmental disorders including ADHD and autism (*Hoffmire et al., 2014*; *Krakowiak et al., 2008*; *Kotagal, 2015*; *Ednick et al., 2009*). Although mechanisms encoding the molecular clock are well understood, little is known about how rhythmic behaviors first emerge (*Vallone et al., 2007*; *Carmona-Alcocer et al., 2018*; *Sehgal et al., 1992*; *Poe et al., 2022*). In particular, cues that trigger the consolidation of sleep and waking behaviors as development proceeds are unclear (*Blumberg et al., 2005*; *Blumberg et al., 2014*; *Frank et al., 2017*; *Frank, 2020*).

**eLife digest** Like most young animals, babies must obtain enough nutrients and energy to grow, yet they also need to rest for their brains to mature properly. As many exhausted new parents know first-hand, balancing these conflicting needs results in frequent, rapid switches between eating and sleeping. Eventually, new-borns' internal biological clock system, which is aligned with the 24-hour light cycle, becomes fully operational. Exactly how this then translates into allowing them to stay alert during the day and be sleepy at night is still unclear.

Like humans, the larvae of fruit flies first sleep haphazardly before developing a circadian pattern whereby they sleep at night and eat during the day. This shift occurs when a group of nerve cells called DN1a, whose job is to 'keep time', connects with Dh44, a subset of neurons which, when active, promote wakefulness. The trigger for these changes, however, has remained elusive.

In response, Poe et al. hypothesized that feeding behaviour and nutrient availability coordinated the emergence of sleep rhythms in fruit flies. Forcing fruit fly larvae to keep feeding in an 'immature' pattern – by either genetic manipulations or reducing the sugar content of their food – not only prevented them from developing 'mature' sleeping rhythms but also resulted in memory problems.

These experiments also showed that the DN1a-Dh44 connection depends on nutrient availability, as it did not form in larvae raised on the low-sugar food. Further genetic experiments showed that the Dh44 cells themselves act like nutrient sensors during the emergence of sleeping patterns.

These results shed new light on the factors triggering sleep rhythm development. Poe et al. hope that the understanding gained can be extended to humans and eventually help manage nervous system disorders and health problems associated with disrupted sleep during early life.

A key potential factor in the maturation of sleep patterns is the coincident change in feeding and metabolism during development. Early in development, most young animals must obtain enough nutrients to ensure proper growth (*Ghosh et al., 2023*). Yet, developing organisms must also sleep to support nervous system development (*Vallone et al., 2007*). These conflicting needs (feeding vs. quiescence) result in rapid transitions between sleeping and feeding states early in life (*Adair and Bauchner, 1993*; *Levin and Stern, 1975*). As development proceeds, nutritional intake and storage capacity increase, allowing for the consolidation of feeding and sleep to specific times of day (*Li et al., 2018*). These changes in nutritional storage capacity are likely conserved as mammalian body composition and nutritional capacity change over infant development (*Toro-Ramos et al., 2015*) and *Drosophila* show rapid increases in overall larval body size including the size of the fat body (used for nutrient storage) across development (*Butterworth et al., 1988*; *Jacobs et al., 2020*). However, the role that developmental change in metabolic drive plays in regulating the consolidation of behavioral rhythms is not known.

In adult *Drosophila*, sleep and feeding behaviors are consolidated to specific times of day with flies eating more in the day than the night (*Barber et al., 2016*). Yet, early in development, sleep in 2nd instar *Drosophila* larvae (L2) lacks a circadian pattern (*Poe et al., 2023*). We previously determined that sleep-wake rhythms are initiated in early 3rd instar *Drosophila* larvae (L3) (72 hr AEL). DN1a clock neurons anatomically and functionally connect to Dh44 arousal output neurons to drive the consolidation of sleep in L3. Development of this circuit promotes deeper sleep in L3 resulting in the emergence of long-term memory (LTM) capabilities at the L3 stage but not before (*Poe et al., 2023*).

Here, we identify the cues that trigger the emergence of the DN1a-Dh44 circuit and the consolidation of sleep-wake rhythms in *Drosophila* larvae. We demonstrate that developmental changes to energetic demands drive consolidation of periods of sleep and feeding across the day as animals mature. While endogenous deeper sleep in L3 facilitates LTM, we find that experimentally inducing deep sleep prematurely in L2 is detrimental to development and does not improve LTM performance. Additionally, we demonstrate that DN1a-Dh44 circuit formation is developmentally plastic, as rearing on an insufficient nutritional environment prevents establishment of this neural connection. Finally, we find that Dh44 neurons require glucose metabolic genes to promote sleep-wake rhythm development, suggesting that these neurons sense the nutritional environment to promote circadian-sleep crosstalk.

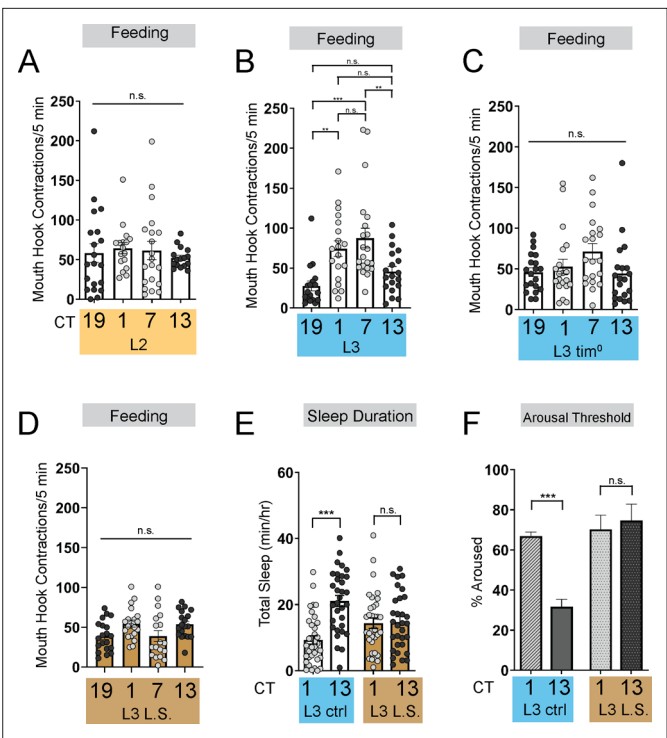

**Figure 1.** Energetic drive limits sleep rhythm development. (**A–D**) Feeding rate (# of mouth hook contractions per 5 min) of L2 controls (**A**) L3 raised on regular (ctrl) food (**B**) L3 clock mutants (**C**) and L3 raised on low sugar (L.S.) food (**D**) across the day. (**E**) Sleep duration at CT1 and CT13 in L3 raised on regular (ctrl) and L.S. food. (**F**) Arousal threshold at CT1 and CT13 in L3 raised on regular (ctrl) and L.S. food. A-D, n=18–20 larvae; E, n=29–34 larvae; (**F**) n=100–172 sleep episodes, 18 larvae per genotype. One-way ANOVAs followed by Sidak's multiple comparisons tests (**A–D**) Two-way ANOVAs followed by Sidak's multiple comparison test (**E–F**). For this and all other figures unless otherwise specified, data are presented as mean ± SEM; n.s., not significant, *p<0.05, **p<0.01, ***p<0.001. Source data in *Figure 1—source data 1*.

The online version of this article includes the following source data and figure supplement(s) for figure 1:

**Source data 1.** Raw values for sleep and feeding in genetic and nutritional manipulations.

**Figure supplement 1.** Larvae reared on low sugar diet develop normally.

**Figure supplement 1—source data 1.** Measures of all sleep and developmental parameters on L.S. food.

## Results

### Energetic demands limit developmental onset of rhythmic behaviors

The emergence of circadian sleep-wake rhythms in *Drosophila* larvae is advantageous as it enables long-term memory capabilities at the L3 stage (*Poe et al., 2023*). Less mature larvae (L2) do not exhibit consolidated sleep-wake patterns, prompting us to ask why rhythmic behaviors do not emerge earlier in life. To determine if the absence of rhythmic sleep-wake patterns in L2 might be related to feeding patterns, we examined larval feeding rate (# mouth hook contractions in a 5-min period) under constant conditions at 4 times across the day (Circadian Time [CT] 1, CT7, CT13, and CT19) in developmentally age-matched 2nd (L2) and early 3rd instar (L3; 72 hr AEL) larvae. While we observed no differences in feeding rate across the day in L2, we found that L3 show diurnal differences in feeding with higher feeding rate during the subjective day compared to the subjective night (*Figure 1A and B*). Analysis of total food intake indicated that L2 consume the same amount at CT1 and CT13; however, L3 consume more at CT1 than at CT13 (*Figure 1—figure supplement 1A*). To assess whether the daily pattern of feeding in early L3 is dependent on the canonical circadian biological clock, we examined feeding rate in null mutants for the clock gene *tim* (*Konopka and Benzer, 1971*). We observed no differences in feeding rate across the day in L3 *tim* mutants indicating that the daily feeding pattern requires a functioning molecular clock (*Figure 1C*). These findings underscore

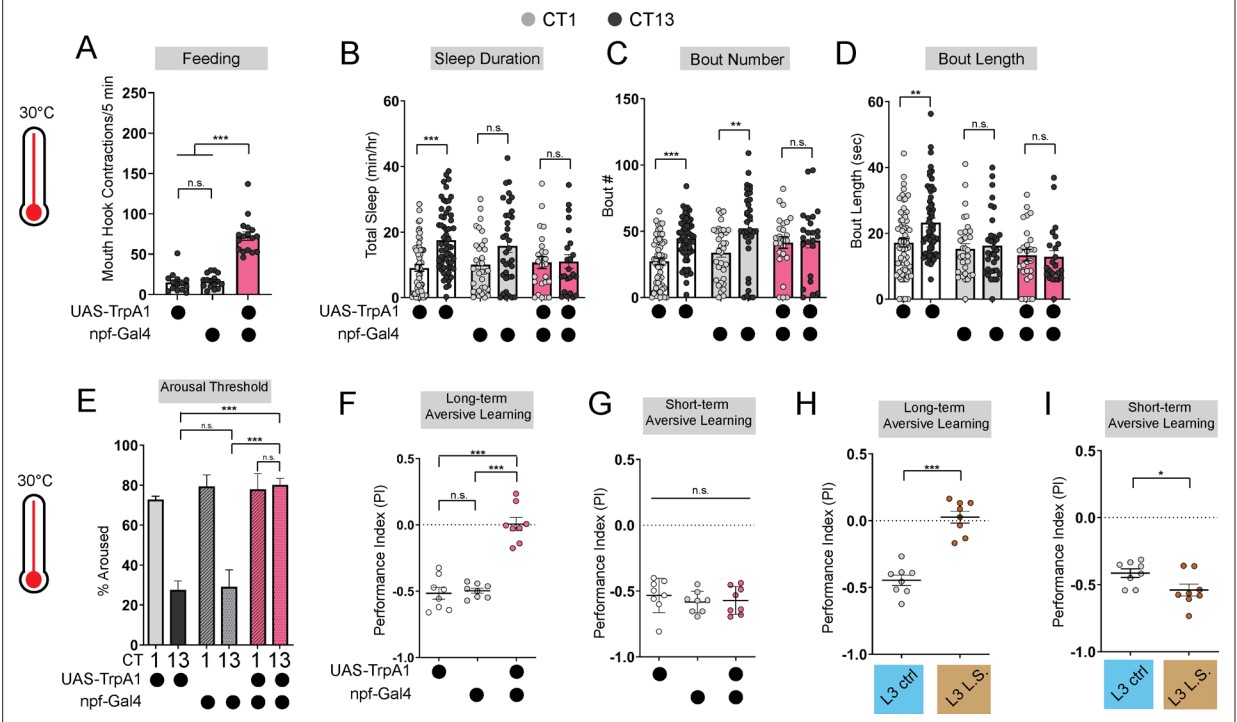

**Figure 2.** Immature feeding strategies limit LTM. (**A**) Feeding rate of L3 expressing *npf*-Gal4 >*UAS-TrpA1* and genetic controls at 30 °C at CT13. (**B–D**) Sleep duration (**B**), bout number (**C**), and bout length (**D**) of L3 expressing *npf*-Gal4 >*UAS-TrpA1* and genetic controls at 30 °C. (**E–G**) Arousal threshold (**E**), long-term aversive memory performance (**F**), and short-term aversive memory performance (**G**) in L3 expressing *npf*-Gal4 >*UAS-TrpA1* and genetic controls at 30 °C. (**H,I**) Short-and long-term term aversive memory performance in L3 raised on ctrl and L.S. food. A, n=18–20 larvae; B-D, n=24–61 larvae; E, n=125–160 sleep episodes, 18 larvae per genotype; F-I, n=8 PIs (240 larvae) per genotype. One-way ANOVAs followed by Sidak's multiple comparisons tests (**A**) and (**E–G**) Two-way ANOVAs followed by Sidak's multiple comparison test (**B–D**) unpaired two-tailed Student's *t*-test (**H and I**). Source data in *Figure 2—source data 1*.

The online version of this article includes the following source data and figure supplement(s) for figure 2:

**Source data 1.** Sleep and LTM values for npf manipulations at 30c.

**Figure supplement 1.** Baseline odor preferences, feeding, and sleep are not affected by *npf*-Gal4 manipulations.

**Figure supplement 1—source data 1.** Temperature control data for npf manipulations.

the tight relationship between sleep and feeding across development as diurnal differences in sleep emerge concurrently at the L3 stage (*Poe et al., 2023*).

To further investigate how the emergence of circadian sleep is related to changes in feeding patterns during development, we asked whether enforcing a constant (immature) feeding pattern at the L3 stage affects sleep-wake rhythms. First, we devised a nutritional paradigm with reduced sugar content but otherwise normal food (low sugar, 1.2% glucose, L.S.). Critically, this paradigm did not affect any measures of larval growth or development (*Figure 1—figure supplement 1D–F*) in contrast to numerous other diets that were assessed. We found that the feeding pattern in L3 raised on L.S. food closely resembled that of L2 on normal food (8% glucose), with feeding spread out across the day (*Figure 1D*; *Figure 1—figure supplement 1A*). Compared to L3 raised on control food, this paradigm was also associated with loss of diurnal differences in sleep duration, sleep bout number, and arousal threshold (indicating less deep sleep) (*Figure 1E and F*; *Figure 1—figure supplement 1B and C*). Next, to avoid chronic effects of this dietary manipulation, we acutely stimulated feeding in L3 reared on normal food using thermogenetic activation of NPF +neurons (*Figure 2A*; *Figure 2—figure supplement 1A*; *Wu et al., 2003*; *Chung et al., 2017*). Like L.S. conditions, enforcing a constant feeding pattern through NPF +neuron activation led to loss of sleep rhythms and loss of deep sleep (higher arousal threshold) in L3 (*Figure 2B–E*; *Figure 2—figure supplement 1B–E*).

Disruption to circadian sleep and/or to deeper sleep stages during development is associated with impairments in long-term memory formation (*Ameen et al., 2022*; *Konrad et al., 2016*; *Seehagen*

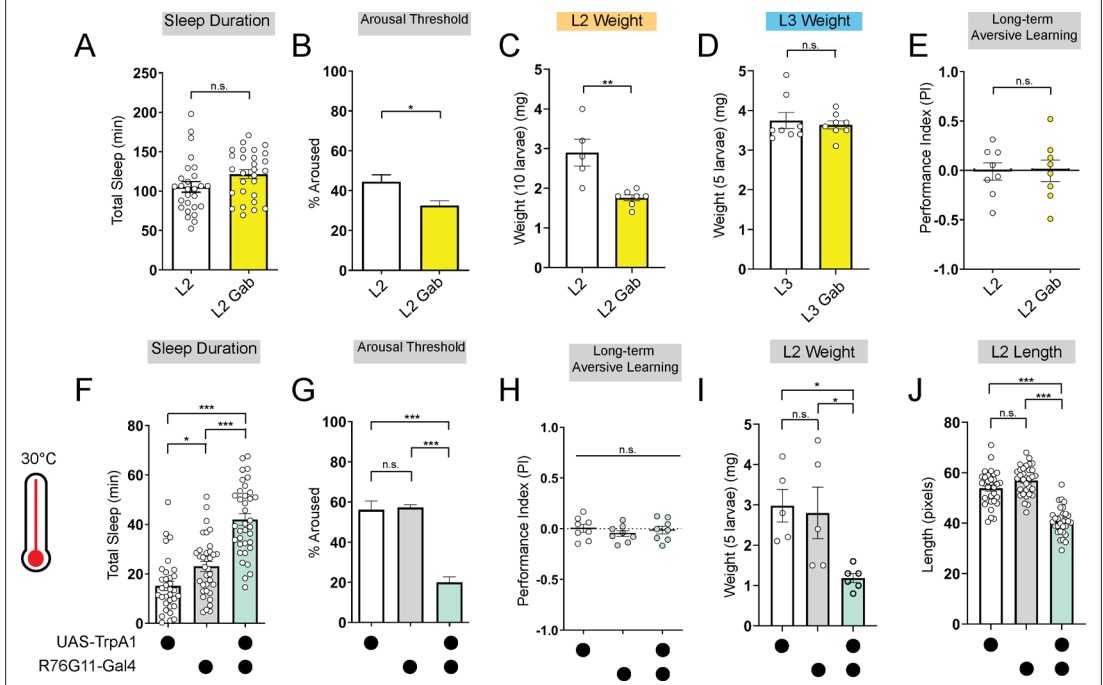

**Figure 3.** Deeper sleep in L2 is energetically disadvantageous. (**A, B**) Sleep duration (**A**) and arousal threshold (**B**) of L2 control fed vehicle control (**L2**) or Gaboxadol (L2 Gab). (**C, D**) Total body weight of L2 (**C**) (in groups of 10) or L3 (**D**) (in groups of 5) fed vehicle control or Gaboxadol (Gab). (**E**) Long-term aversive memory performance in L2 fed vehicle control (**L2**) or Gaboxadol (L2 Gab). (**F, G**) Sleep duration (**F**) and arousal threshold (**G**) of L2 expressing *R76G11*-Gal4>*UAS-TrpA1* and genetic controls at 30 °C. (**H**) Long-term aversive memory performance of L2 expressing *R76G11*-Gal4>*UAS-TrpA1* and genetic controls at 30 °C. (**I, J**) Total body weight (**I**) and total body length (**J**) of L2 expressing *R76G11*-Gal4>*UAS-TrpA1* and genetic controls at 30 °C. (**A**) n=28 larvae; (**B**) n=110–220 sleep episodes, 18 larvae per genotype; (**C**) n=5–7 groups (50–70 larvae); (**D**) n=8 groups (40 larvae); (**E**) n=8 PIs (240 larvae) per genotype; (**F**) n=33–36 larvae; (**G**) n=234–404 sleep episodes, 30–40 larvae per genotype; (**H**) n=8 PIs (240 larvae) per genotype; (**I**) n=5 groups (25 larvae); (**J**) n=31–32 larvae. Unpaired two-tailed Student's *t*-tests (**A–E**) one-way ANOVAs followed by Sidak's multiple comparisons tests (**F–J**). Source data in *Figure 3—source data 1*.

The online version of this article includes the following source data and figure supplement(s) for figure 3:

**Source data 1.** Sleep, LTM, and developmental data for L2 manipulations.

**Figure supplement 1.** Baseline sleep and odor preferences are not disrupted by *R76G11*-Gal4 manipulations.

**Figure supplement 1—source data 1.** Temperature control and naive preference data for L2 manipulations.

---

*et al., 2015*; *Jones et al., 2019*; *Smarr et al., 2017*). We next asked if the loss of deeper sleep observed in animals adopting an immature (constant) feeding strategy through either the dietary paradigm or NPF +neuron activation affects long-term memory (LTM). Consistent with deeper sleep stages being necessary for LTM, we observed a loss of LTM in either the NPF +neuron activation (*Figure 2F*; *Figure 2—figure supplement 1F–I*) or under L.S. conditions (*Figure 2H*; *Figure 1—figure supplement 1G–I*); short-term memory (STM) was intact in both manipulations (*Figure 2G and I*). These findings suggest that immature feeding strategies preclude the emergence of sleep rhythms and LTM. Together, our data indicate that consolidated periods of sleep and feeding emerge due to developmentally dynamic changes in energetic demands.

## Deeper sleep stages are energetically disadvantageous in L2

To investigate if promoting deep sleep at night in L2 can enable precocious LTM abilities, we fed L2 larvae the GABA-A agonist gaboxadol (*Dissel et al., 2015*; *Dijk et al., 2012*). Gaboxadol feeding induced deeper sleep in L2 (as reflected by an increase in arousal threshold) although sleep duration was unchanged. Despite achieving deeper sleep, LTM was still not evident in L2 (*Figure 3A, B and E*; *Figure 3—figure supplement 1A and B*); however, in contrast to L3, L2 on gaboxadol failed to develop normally (*Figure 3C and D*). Next, to avoid chronic pharmacological manipulations altogether, we acutely stimulated sleep-inducing neurons using thermogenetic approaches. We found that acute

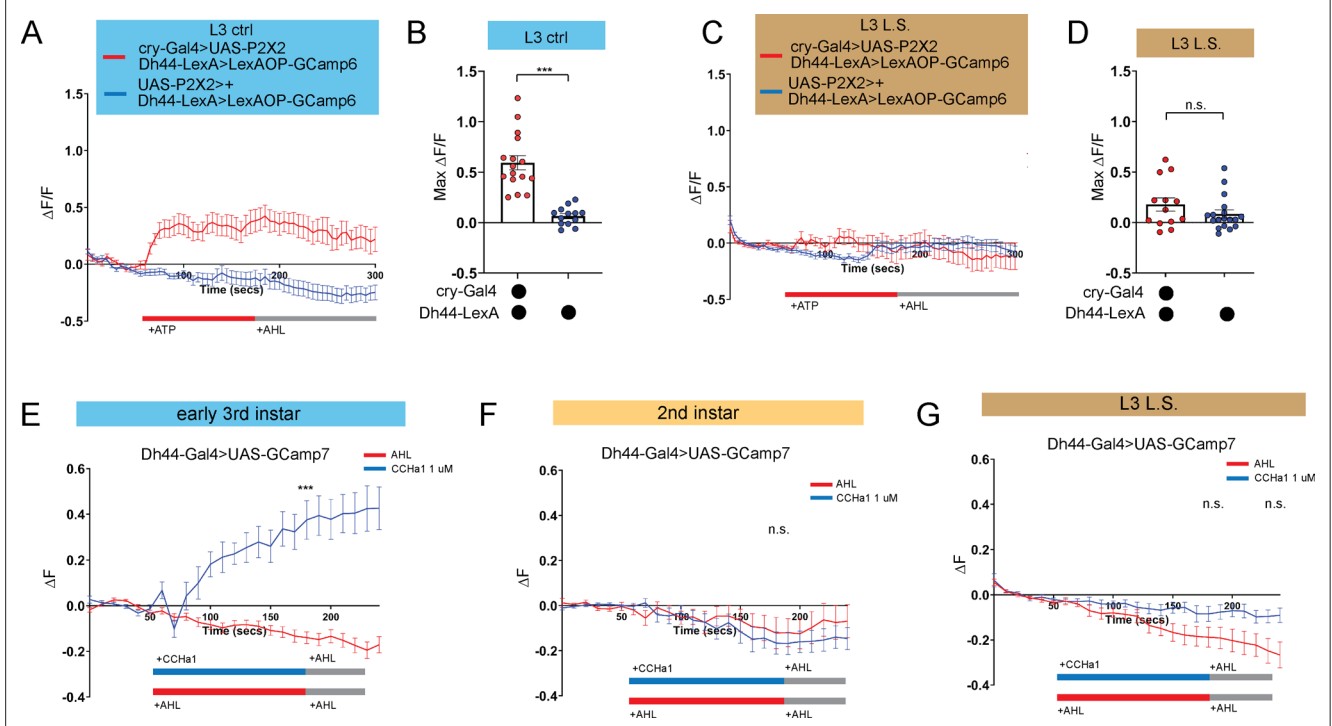

**Figure 4.** DN1a-Dh44 circuit formation is developmentally plastic. (**A, C**) GCaMP6 signal in Dh44 neurons with activation of DN1a neurons in L3 controls (**A**) and L3 raised on L.S. food (**C**). Red bar indicates ATP application and gray bar indicates AHL application. (**B, D**) Maximum GCaMP change (ΔF/F) for individual cells in L3 controls (**B**) and L3 raised on L.S. food (**D**). (**E–G**) GCaMP7 signal in Dh44 neurons during bath application of 1 μM CCHa1 synthetic peptide in L3 controls (**E**), L2 controls (**F**) and L3 raised on L.S. food (**G**) brains. Red/blue bar indicates timing of CCHa1 (blue) or buffer (AHL, red) application and gray bar indicates timing of washout AHL application. (**A-D**) n=12–18 cells, 8–10 brains; (**E-G**) n=11–15 cells, 5–10 brains. Unpaired two-tailed Student's *t*-tests (**B and D**) Mann-Whitney U test (**E–G**). Source data in *Figure 4—source data 1*.

The online version of this article includes the following source data for figure 4:

**Source data 1.** Individual cell fluorescence data for P2X2 and CCHa1 imaging.

activation of these neurons caused an increase in sleep duration and bout length with a decrease in arousal threshold (*Figure 3F and G*; *Figure 3—figure supplement 1C–H*). However, inducing deeper sleep in L2 via genetic approaches likewise did not improve LTM performance (*Figure 3H*; *Figure 3—figure supplement 1I–K*) despite STM being intact (*Figure 3—figure supplement 1L*). Moreover, as with gaboxadol feeding in L2, thermogenetic induction of deeper sleep stages disrupted larval development (*Figure 3I and J*). These data suggest that sleep cannot be leveraged to enhance cognitive function prematurely because prolonged periods of deep sleep are not energetically sustainable at this stage.

## DN1a-Dh44 circuit formation is developmentally plastic

We previously determined that Dh44 arousal neurons anatomically and functionally connect to DN1a clock neurons at the L3 stage (*Poe et al., 2023*). Therefore, we examined the functional connectivity between clock and arousal loci in the setting of nutritional perturbation by expressing ATP-gated P2X2 receptors *Yao et al., 2012* in DN1a neurons and GCaMP6 in Dh44 neurons. As expected, activation of DN1as in L3 raised on control food caused an increase in calcium in Dh44 neurons (*Figure 4A and B*; *Poe et al., 2023*). In contrast, in L3 raised on L.S. conditions, activation of DN1as no longer elicited a response in Dh44 neurons (*Figure 4C and D*). Thus, the nascent connection underlying circadian sleep in *Drosophila* is developmentally plastic: in an insufficient nutritional environment, this connection is not functional, presumably to facilitate a more constant feeding pattern that fulfills energetic needs of the animal. However, this feeding pattern eschews deep sleep at the expense of LTM.

We previously determined that release of *CCHamide-1* (*CCHa1*) from DN1as to *CCHa1mide-1 receptor* (*CCHa1-R*) on Dh44 neurons is necessary for sleep-wake rhythms in L3 (*Poe et al., 2023*).

Our data support a model in which the downstream Dh44 neurons are poised to receive clock (DN1a) input as soon the connection forms between these cellular populations. To test this idea directly, we next asked whether Dh44 neurons in L2, before the DN1a-Dh44 connection has formed, are competent to receive the CCHa1 signal. CCHamide-1 peptide was bath applied onto dissected larval brains expressing *UAS-GCaMP7* in Dh44 neurons. As anticipated, we observed an increase in intracellular calcium in early L3 (*Figure 4E*); surprisingly, CCHa1 application did not alter calcium levels in Dh44 neurons in L2 (*Figure 4F*), indicating that Dh44 neurons are not capable of receiving CCHa1 input prior to early L3. Moreover, CCHa1 application in L3 reared on L.S. also failed to elicit a response in Dh44 neurons (*Figure 4G*) suggesting that sub-optimal nutritional milieu influences the development of Dh44 neuronal competency to receive clock-driven cues. Thus, our data indicate that the nutritional environment influences DN1a-Dh44 circuit development.

## Dh44 neurons require glucose metabolic genes to regulate sleep-wake rhythms

How do developing larvae detect changes in their nutritional environments to drive the circadian consolidation of sleep and feeding at the L3 stage? In adult *Drosophila*, Dh44 neurons act as nutrient sensors to regulate food consumption and starvation-induced sleep suppression through the activity of both glucose and amino acid sensing genes (*Dus et al., 2015*; *Yang et al., 2018*; *Oh et al., 2021*; *Oh and Suh, 2023*). Indeed, Dh44 neurons themselves are activated by changes in bath application of nutritive sugars (*Dus et al., 2015*). We next asked whether Dh44 neurons in L3 require metabolic genes to regulate sleep-wake rhythms. In L3 raised on regular food, we conducted an RNAi-based candidate screen of different glucose and amino acid sensing genes known to act in adult Dh44 neurons. Sleep duration at CT1 and CT13 was assessed with knockdown of glucose metabolic genes (*Hexokinase-C*, *Glucose transporter 1*, and *Pyruvate kinase*) or amino acid sensing genes (*Gcn2* and its downstream target ATF4 or *cryptocephal*) in Dh44 neurons. We found that knockdown of glucose metabolism genes, *Glut1*, *Hex-C* and *PyK*, in Dh44 neurons resulted in a loss of rhythmic changes in sleep duration and bout number (*Figure 5A–C*; *Figure 5—figure supplement 1A–F*) with no effect on L3 feeding in the *Hex-C* manipulation (*Figure 5D*). Additionally, knockdown of *Hex-C* in DN1as did not disrupt rhythmic changes in sleep duration in L3 (*Figure 5—figure supplement 1G–I*) suggesting a specialized role for glucose metabolism in Dh44 neurons for sleep-wake rhythm maturation. Manipulation of glucose metabolic genes in L2 did not affect sleep duration at CT1 and CT13 (*Figure 5E–G*; *Figure 5—figure supplement 1J–O*) providing evidence that nutrient sensing is not required at this stage to regulate sleep. In contrast to their role in adult Dh44 neurons, knockdown of amino acid sensing genes, *Gcn2* and *crc*, in Dh44 neurons did not disrupt rhythmic changes in sleep duration and bout number in L3 (*Figure 5H and I*; *Figure 5—figure supplement 2A–D*) suggesting that Dh44 neurons may not act through amino acid sensing pathways to regulate sleep-wake rhythm development. Thus, Dh44 neurons require glucose metabolic genes to drive sleep-wake rhythm development. Our data indicate that the emergence of daily sleep-wake patterns is regulated by developmental changes in energetic capacity, and suggest that Dh44 neurons may be necessary for sensing of larval nutritional environments.

## Discussion

Nutritional environment and energetic status exert profound effects on sleep patterns during development, but mechanisms coupling sleep to these factors remain undefined. We report that the development of sleep-circadian circuits depends on organisms achieving sufficient nutritional status to support the consolidation of deep sleep at night (*Figure 5—figure supplement 3*). Our data demonstrate that larval Dh44 neurons require glucose metabolic genes but not amino acid sensing genes to modulate sleep-wake rhythms. Larval Dh44 neurons may therefore have distinct functions from their adult counterparts for integrating information about the nutritional environment through the direct sensing of glucose levels to modulate sleep-wake rhythm development. Maintaining energy homeostasis and sensing of the nutritional environment are likely conserved regulators of sleep-wake rhythm development, as young mice exposed to a maternal low-protein diet show disruptions in night-time sleep architecture and energy expenditure later in life (*Sutton et al., 2010*; *Crossland et al., 2017*).

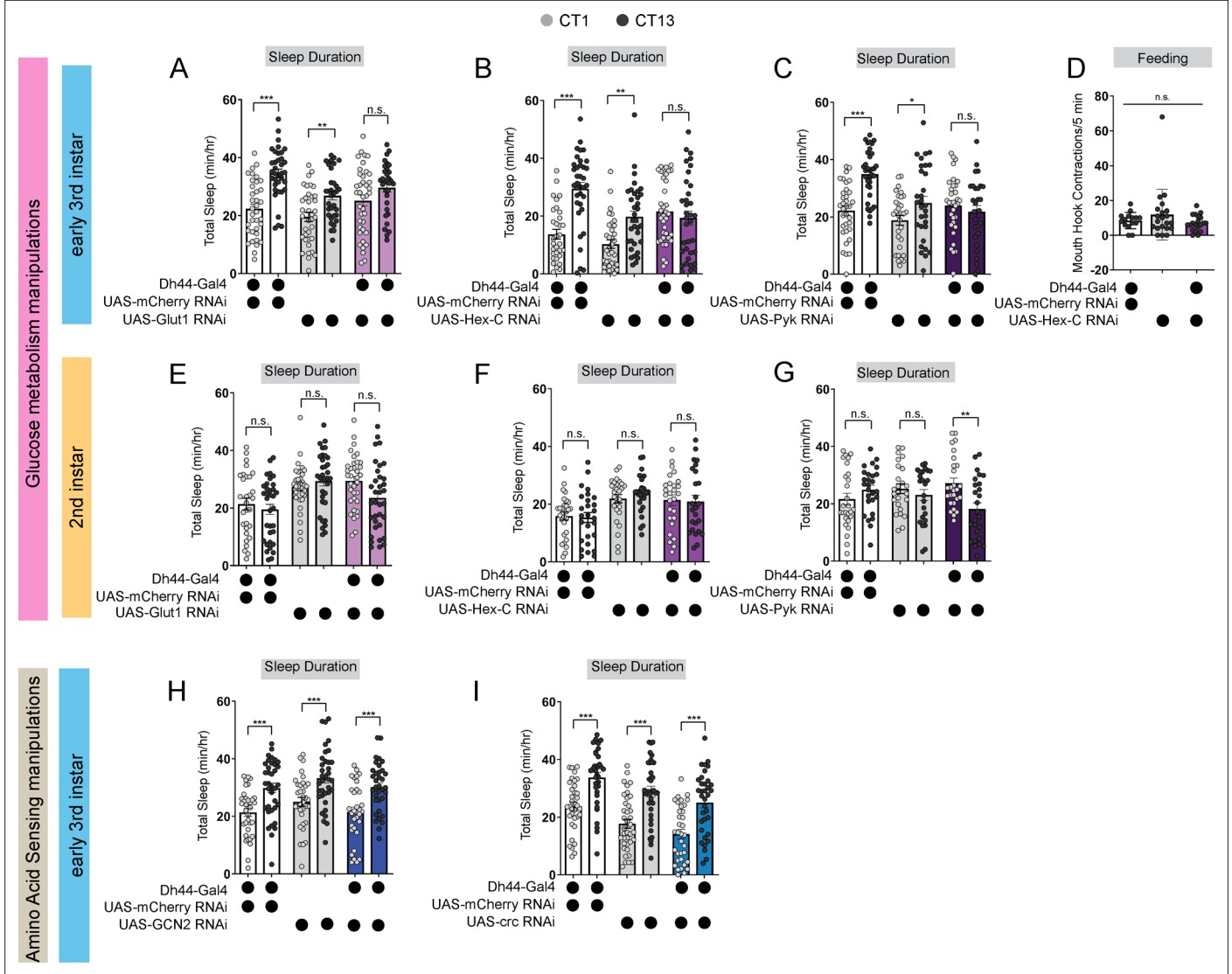

**Figure 5.** Dh44 neurons require glucose metabolic genes to regulate sleep-wake rhythms. (**A–C**) Sleep duration in L3 expressing *UAS-Glut1-RNAi* (**A**), *UAS-Hex-C-RNAi* (**B**), and *UAS-Pyk-RNAi* (**C**) with *Dh44-Gal4* and genetic controls at CT1 and CT13. (**D**) Feeding rate (# of mouth hook contractions per 5 min) of L3 expressing *UAS-Hex-C-RNAi* with *Dh44-Gal4* and genetic controls at CT13. (**E–G**) Sleep duration in L2 expressing *UAS-Glut1-RNAi* (**E**), *UAS-Hex-C-RNAi* (**F**), and *UAS-Pyk-RNAi* (**G**) with *Dh44-Gal4* and genetic controls at CT1 and CT13. (**H, I**) Sleep duration in L3 expressing *UAS-GCN2-RNAi* (**H**) and *UAS-crc-RNAi* (**I**) with *Dh44-Gal4* and genetic controls at CT1 and CT13. (**A-C**) n=32–40 larvae; (**D**) n=20 larvae; (**E-I**) n=32–40 larvae. Two-way ANOVAs followed by Sidak's multiple comparison test (**A–C**) and (**E–I**) One-way ANOVAs followed by Sidak's multiple comparisons tests (**D**). Source data in *Figure 5—source data 1*.

The online version of this article includes the following source data and figure supplement(s) for figure 5:

**Source data 1.** Total sleep duration measures for Dh44 metabolism manipulations in L2 and L3.

**Figure supplement 1.** Glucose metabolic gene manipulations affect L3 sleep.

**Figure supplement 1—source data 1.** Complete sleep data for Dh44 glucose metabolism manipulations.

**Figure supplement 2.** Amino acid sensing gene manipulations do not affect L3 sleep.

**Figure supplement 2—source data 1.** Raw values for amino acid sensing manipulations in Dh44 neurons.

**Figure supplement 3.** Model Figure.

Together with our previously published work, our findings support a model in which changes in both overall circuit development and molecular changes in post-synaptic (Dh44) neurons likely drive sleep-wake rhythm circuit development. Our CCHa1 peptide data suggest that Dh44 neurons may undergo changes in CCHa1-R expression or subcellular localization between the L2 and L3 stages:

we only observed an increase in Dh44 neural activity in response to the bath application of CCHa1 peptide at the L3 stage (*Figure 4E and F*). Interestingly, this increase in activity is absent in low nutrient conditions (*Figure 4G*) suggesting that the larval nutritional environment may also modulate CCHa1-R localization or expression. Indeed, we observed a disruption of sleep rhythms in L3 when glucose metabolic genes are knocked down in Dh44 neurons, demonstrating that post-synaptic processes likely initiate onset of circadian sleep. These findings raise intriguing questions for how changes in an organism's energetic and nutritional state influence sleep-circadian circuit development. Perhaps larval Dh44 neurons respond to an increase in glucose levels in the environment by promoting CCHa1-R localization to the membrane. In this model, changes in CCHa1-R subcellular localization allow Dh44 neurons to become competent to receive clock-driven cues, while this or other Dh44-derived signals promote circuit connectivity with DN1as to drive consolidation of sleep at the L3 stage. There are no available antibodies or endogenous fluorescent reporters of CCHa1-R, limiting our ability to examine receptor subcellular localization. Additionally, while our study focuses on presumed CCHa1 synaptic signaling between DN1a and Dh44 neurons, we cannot rule out the possibility of CCHa1 volume transmission from DN1as or other sources as contributors to sleep-wake regulation. Regardless, our data open avenues for future work on the molecular and subcellular mechanisms regulating DN1a-Dh44 circuit development.

Our findings demonstrate that larvae exhibit both sleep-wake and feeding daily rhythms at the L3 stage, but not earlier (*Poe et al., 2023*). This raises the obvious question of whether sleep and feeding are opposite sides of the same coin. While larvae cannot eat when they are sleeping, we have observed distinct effects of certain manipulations on sleep behaviors but not feeding. For example, knockdown of *Hex-C* in Dh44 neurons disrupts sleep rhythms with no obvious effect on feeding behavior (*Figure 5D*). It is, of course, possible to affect both sleep and feeding behaviors with the same manipulation (e.g. activation of NPF +neurons) underscoring that they are highly interconnected behaviors. Future work will leverage the larval system to examine how sleep-wake and feeding circuitry communicate to balance these rhythmic behaviors across developmental periods.

## Materials and methods

### Fly stocks

The following lines have been maintained as lab stocks or were obtained from Dr. Amita Sehgal: iso31, tim0 (*Konopka and Benzer, 1971*), Dh44$^{VT}$-Gal4 (VT039046) (*Jenett et al., 2012*), cry-Gal4 pdf-Gal80 (*Stoleru et al., 2004*), UAS-TrpA1 (*Hamada et al., 2008*), UAS-mCherry RNAi, LexAOP-GCaMP6 UAS-P2X2 (*Yao et al., 2012*), and UAS-GCaMP7f. Dh44-LexA (80703), npf-Gal4 (25681), R76G11-Gal4 (48333), Hex-C RNAi (57404), Glut1 RNAi (40904), PyK RNAi (35218), GCN2 RNAi (67215), and crc RNAi (80388) were from the Bloomington *Drosophila* Stock Center (BDSC).

### Larval rearing and sleep assays

Larval sleep experiments were performed as described previously (*Poe et al., 2023*; *Szuperak et al., 2018*). Briefly, molting 2$^{nd}$ instar or 3$^{rd}$ instar larvae were placed into individual wells of the LarvaLodge containing either 120 µl (for L2) or 95 µl (for L3) of 3% agar and 2% sucrose media covered with a thin layer of yeast paste. The LarvaLodge was covered with a transparent acrylic sheet and placed into a DigiTherm (Tritech Research) incubator at 25 °C for imaging. Experiments were performed in the dark. For thermogenetic experiments, adult flies were maintained at 22 °C. Larvae were then placed into the LarvaLodge (as described above) which was moved into a DigiTherm (Tritech Research) incubator at 30 °C for imaging.

### LarvaLodge image acquisition and processing

Images were acquired every 6 s with an Imaging Source DMK 23GP031 camera (2592 X 1944 pixels, The Imaging Source, USA) equipped with a Fujinon lens (HF12.55A-1, 1:1.4/12.5 mm, Fujifilm Corp., Japan) with a Hoya 49mm R72 Infrared Filter as described previously (*Poe et al., 2023*; *Szuperak et al., 2018*). We used IC Capture (The Imaging Source) to acquire time-lapse images. All experiments were carried out in the dark using infrared LED strips (Ledlightsworld LTD, 850 nm wavelength) positioned below the LarvaLodge.

Images were analyzed using custom-written MATLAB software (see *Churgin et al., 2019* and *Szuperak et al., 2018*) (Code available in Supplement). Temporally adjacent images were subtracted to generate maps of pixel value intensity change. A binary threshold was set such that individual pixel intensity changes that fell below 40 gray-scale units within each well were set equal to zero ('no change') to eliminate noise. For 3rd instars, the threshold was set to 45 to account for larger body size. Pixel changes greater than or equal to threshold value were set equal to one ('change'). Activity was then calculated by taking the sum of all pixels changed between images. Sleep was defined as an activity value of zero between frames. For 2nd instar sleep experiments done across the day, total sleep was summed over 6 hr beginning 2 hr after the molt to second instar. For sleep experiments performed at certain circadian times, total sleep in the second hour after the molt to second (or third) instar was summed.

## Feeding behavior analysis

For feeding rate analysis, newly molted 2nd instar or 3rd instar larvae were placed in individual wells of the LarvaLodge containing 120 µl of 3% agar and 2% sucrose media covered with a thin layer of yeast paste. Larvae were then imaged continuously with a Sony HDR-CX405 HD Handycam camera (B&H Photo, Cat. No: SOHDRCX405) for 5 minutes. The number of mouth hook contractions (feeding rate) was counted manually over the imaging period and raw numbers were recorded. For food intake analysis, newly molted 2nd instar or 3rd instar larvae were starved for 1 hr in petri dishes with water placed on a Kimwipe. To compare groups of larvae of similar body weights, 13 L3 larvae and 26 L2 larvae were grouped together. Larvae were placed in a petri dish containing blue-dyed 3% agar, 2% sucrose, and 2.5% apple juice with blue-dyed yeast paste on top for 4 hr at 25 °C in constant darkness. We found that 4 hr on blue-dyed agar was sufficient to reflect total food consumption in each condition with a shorter period of time (1 hr) causing more variability. After 4 hr, groups of larvae were washed in water, put in microtubes, and frozen at –80 °C for 1 hr. Frozen larvae were then homogenized in 300 µl of distilled water and spun down for 5 min at 13,0000 rpm. The amount of blue dye in the supernatant was then measured using a spectrophotometer ($OD_{629}$). Food intake represents the OD value of each measurement.

## Aversive olfactory conditioning

We used an established two odor reciprocal olfactory conditioning paradigm with 10 mM quinine (quinine hydrochloride, EMSCO/Fisher, Cat. No: 18-613-007) as a negative reinforcement to test short-term or long-term memory performance in L2 and early L3 larvae (*Widmann et al., 2016*) at CT12-15 (*Poe et al., 2023*). Experiments were conducted on assay plates (100X15 mm, Genesee Scientific, Cat. No: 32–107) filled with a thin layer of 2.5% agarose containing either pure agarose (EMSCO/Fisher, Cat. No: 16500–500) or agarose plus reinforcer. As olfactory stimuli, we used 10 µl amyl acetate (AM, Sigma-Aldrich, Cat. No: STBF2370V, diluted 1:50 in paraffin oil-Sigma-Aldrich, Cat. No: SZBF140V) and octanol (OCT, Fisher Scientific, Cat. No: SALP564726, undiluted). Odorants were loaded into the caps of 0.6 mL tubes (EMSCO/Fisher, Cat. No: 05-408-123) and covered with parafilm (EMSCO/Fisher, Cat. No: 1337412). For naïve preferences of odorants, a single odorant was placed on one side of an agarose plate with no odorant on the other side. A group of 30 larvae were placed in the middle. After 5 min, individuals were counted on the odorant side, the non-odorant side, or in the middle. The naïve preference was calculated by subtracting the number of larvae on the non-odorant side from the number of larvae on the odorant side and then dividing by the total number of larvae. For naïve preference of quinine, a group of 30 larvae were placed in the middle of a half agarose-half quinine plate. After 5 min, individuals were counted on the quinine side, the agarose side, or in the middle. The naïve preference for quinine was calculated by subtracting the number of larvae on the quinine side from the number of larvae on the agarose side and then dividing by the total number of larvae. Larvae were trained by exposing a group of 30 larvae to AM while crawling on agarose medium plus quinine reinforcer. After 5 min, larvae were transferred to a fresh Petri dish containing agarose alone with OCT as an odorant (AM+/OCT). A second group of 30 larvae received the reciprocal training (AM/OCT+). Three training cycles were used for all experiments. For long-term memory, larvae were transferred after training onto agarose plates with a small piece of Kimwipe moistened with tap water and covered in dry active yeast (LabScientific, Cat. No: FLY804020F). Larvae were then kept in the dark for 1.5 hr before testing memory performance. Training and retention for thermogenetic experiments

were conducted at 30 °C. For short-term memory, larvae were immediately transferred after training onto test plates (agarose plus reinforcer) on which AM and OCT were presented on opposite sides of the plate. After 5 min, individuals were counted on the AM side, the OCT side, or in the middle. We then calculated a preference index (PREF) for each training group by subtracting the number of larvae on the conditioned stimulus side from the number of larvae on the unconditioned stimulus side. For one set of experiments, we calculated two PREF values: (1 a) $PREF_{AM+/OCT} = (\#AM - \#OCT)/ \# TOTAL$; (1b) $PREF_{AM/OCT+} = (\#OCT-\#AM)/ \# TOTAL$. We then took the average of each PREF value to calculate an associative performance index (PI) as a measure of associative learning. $PI = (PREF_{AM+/OCT} + PREF_{AM/OCT+})/2$.

## Arousal threshold

Blue light stimulation was delivered as described in *Poe et al., 2023*; *Szuperak et al., 2018* using 2 high power LEDs (Luminus Phatlight PT-121, 460 nm peak wavelength, Sunnyvale, CA) secured to an aluminum heat sink. The LEDs were driven at a current of 0.1 A (low intensity). We used a low intensity stimulus for 4 s every 2 min for 1 hr beginning the 2nd hr after the molt to second (or third) instar. We then counted the number of larvae that showed an activity change in response to stimulus. The percentage of animals that moved in response to the stimulus was recorded for each experiment. For each genotype, at least four biological replicates were performed. We then plotted the average percentage across all replicates.

## P2X2 activation and GCaMP imaging

All live imaging experiments (P2X2 and CCHa1 bath application) were performed as described previously (*Poe et al., 2023*). Briefly, brains were dissected in artificial hemolymph (AHL) buffer consisting of (in mM): 108 NaCl, 5 KCl, 2 CaCl2, 8.2 MgCl2, 4 NaHCO3, 1 NaH2PO4-H20, 5 Trehalose, 10 Sucrose, 5 HEPES, pH = 7.5. Brains were placed on a small glass coverslip (Carolina Cover Glasses, Circles, 12 mm, Cat. No: 633029) in a perfusion chamber filled with AHL.

For P2X2 imaging, dissections were performed at CT12-15 and AHL buffer was perfused over the brains for 1 min of baseline GCaMP6 imaging, then ATP was delivered to the chamber by switching the perfusion flow from the channel containing AHL to the channel containing 2.5 mM ATP in AHL, pH 7.5. ATP was perfused for 2 min and then AHL was perfused for 2 min. Twelve-bit images were acquired with a 40 X water immersion objective at 256X256-pixel resolution. Z-stacks were acquired every 5 s for 3 min. Image processing and measurement of fluorescence intensity was performed in ImageJ as described previously (*Poe et al., 2023*). For each cell body, fluorescence traces over time were normalized using this equation: $\Delta F/F = (F_n-F_0)/F_0$, where $F_n$ = fluorescence intensity recorded at time point n, and $F_0$ is the average fluorescence value during the 1 min baseline recording. Maximum GCaMP change ($\Delta F/F$) for individual cells was calculated using this equation: $\Delta F/F_{max} = (F_{max}-F_0)/F_0$, where $F_{max}$ = maximum fluorescence intensity value recorded during ATP application, and $F_0$ is the average fluorescence value during the 1 min baseline recording. All analysis was done blind to experimental condition.

For CCHa1 bath application, dissections were performed at CT12-15 and AHL buffer was perfused over the brains for 1 min of baseline GCaMP7f imaging, then CCHa1 peptide was delivered to the chamber by switching the perfusion flow from the channel containing AHL to the channel containing 1 μM synthetic CCHa1 in AHL, pH 7.5. CCHa1 was perfused for 2 min, followed by a 1 min wash-out with AHL. For the AHL negative control, the perfusion flow was switched from one channel containing AHL to another channel containing AHL. Twelve-bit images were acquired with a 40 X water immersion objective at 256X256-pixel resolution. Z-stacks were acquired every 10 s for 4 min. Image processing and measurement of fluorescence intensity was performed in ImageJ. A max intensity Z-projection of each time step and Smooth thresholding was used for analysis. Image analysis was performed in a similar manner as for the P2X2 experiments. All analyses were done blind to experimental condition.

## Gaboxadol treatment

Early second or third instar larvae were starved for 1 hr and then fed 75 μl of 25 mg/mL Gaboxadol (hydrochloride) (Thomas Scientific, Cat No: C817P41) in diluted yeast solution for 1 hr prior to loading in LarvaLodge containing 120 μl of 3% agar and 2% sucrose media covered with a thin layer of 25 mg/mL Gaboxadol yeast paste. For LTM experiments, starved early second instars were fed 25 mg/mL

Gaboxadol for 1 hr prior to training and maintained on 25 mg/mL Gaboxadol in diluted yeast solution during retention period.

## Dietary manipulations

Fly food was prepared using the following recipes (based on *Poe et al., 2020*):

| Ingredients | Control food (444 mM glucose) | Low sugar (L.S.) (66 mM glucose) |
|---|---|---|
| Distilled H$_2$O | 234 mL | 234 mL |
| Agar | 2 g (10 g/L) | 2 g (10 g/L) |
| Glucose | 20 g | 3 g |
| Inactive yeast | 20 g | 20 g |
| Acid mix (phosphoric acid +propionic acid) | 2 mL | 2 mL |
| Target final solution volume | 250 mL | 250 mL |

Acid Mix was made by preparing Solution A (41.5 ml Phosphoric Acid mixed with 458.5 ml distilled water) and Solution B (418 ml Propionic Acid mixed with 82 ml distilled water) separately and then mixing Solution A and Solution B together.

Adult flies were placed in an embryo collection cage (Genesee Scientific, cat#: 59–100) and eggs were laid on a petri dish containing either control (ctrl) or Low sugar (L.S.) food. Animals developed on this media for three days.

## Larval body weight and length measurements

For weight, groups of five early 3rd instar larvae raised on either control- or low sugar (L.S.)-filled petri dishes were washed in tap water and dried using a Kimwipe. The five larvae were then weighed as a group on a scale and the weight in mg was recorded. For the Gaboxadol experiments, groups of 10 early 2nd instar larvae or groups of five early 3rd instar larvae were weighed. For length, images of individual early 3rd instar larvae in the LarvaLodge were measured in ImageJ (Fiji) using the straight line tool. The total body length was determined in pixels for individual larvae on each condition.

## Statistical analysis

All statistical analysis was done in GraphPad (Prism). For comparisons between two conditions, two-tailed unpaired *t*-tests were used. For comparisons between multiple groups, ordinary one-way ANOVAs followed by Tukey's multiple comparison tests were used. For comparisons between different groups in the same analysis, ordinary one-way ANOVAs followed by Sidak's multiple comparisons tests were used. For comparisons between time and genotype, two-way ANOVAs followed by Sidak's multiple comparisons tests were used. For comparison of GCaMP signal in CCHa1 experiments, Mann-Whitney test was used. $*p<0.05$, $**p<0.01$, $***p<0.001$. Representative confocal images are shown from at least 8–10 independent samples examined in each case.

## Acknowledgements

We thank members of the Kayser Lab, Raizen Lab, and other members of the Penn Chronobiology and Sleep Institute for helpful discussions and input. This work was supported by NIH DP2NS111996, NIH R01NS120979, and a Burroughs Wellcome Career Award for Medical Scientists to MSK; Hartwell Foundation Fellowship to ARP.

## Additional information

### Funding

| Funder | Grant reference number | Author |
|---|---|---|
| National Institute of Neurological Disorders and Stroke | NIH DP2NS111996 | Matthew S Kayser |
| National Institute of Neurological Disorders and Stroke | NIH R01NS120979 | Matthew S Kayser |
| Burroughs Wellcome Fund | Career Award for Medical Scientists | Matthew S Kayser |
| Hartwell Foundation | Fellowship | Amy R Poe |

The funders had no role in study design, data collection and interpretation, or the decision to submit the work for publication.

### Author contributions

Amy R Poe, Conceptualization, Data curation, Formal analysis, Funding acquisition, Investigation, Methodology, Writing - original draft, Writing - review and editing; Lucy Zhu, Formal analysis, Investigation, Methodology, Writing - review and editing; Si Hao Tang, Formal analysis, Investigation, Writing - review and editing; Ella Valencia, Formal analysis, Investigation; Matthew S Kayser, Conceptualization, Supervision, Funding acquisition, Methodology, Writing - original draft, Project administration, Writing - review and editing

### Author ORCIDs

Amy R Poe ⬥ http://orcid.org/0000-0002-1120-1209
Matthew S Kayser ⬥ http://orcid.org/0000-0003-2359-4967

Joint public review: https://doi.org/10.7554/eLife.97256.3.sa1
Author response https://doi.org/10.7554/eLife.97256.3.sa2

## Additional files

### Supplementary files

• MDAR checklist

• Source code 1. Larval sleep analysis codes.

### Data availability

All data generated or analysed during this study are included in the manuscript and supporting files. Source data files have been provided for all Figures.

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
