## [Editor Report · eLife assessment]

This study presents **valuable** findings as it shows that sleep rhythm formation and memory capabilities depend on a balanced and rich diet in fly larvae. The evidence supporting the claims of the authors is **convincing** with rigorous behavioral assays and state-of-the-art genetic manipulations. The work will be of interest to researchers working on sleep and memory.

---

## [Referee Report · Joint public review]

Summary:

This manuscript investigates how energetic demands affect the sleep-wake cycle in *Drosophila* larvae. L2 stage larvae do not show sleep rhythm and long-term memory (LTM), however, L3 larvae do. The authors manipulate food content to provide insufficient nutrition, which leads to more feeding, no LTM, and no sleep even in older larvae. Similarly, activation of NPF neurons suppresses sleep rhythm. Furthermore, they try to induce a sleep-like state using pharmacology or genetic manipulations in L2 larvae, which can mimic some of the L3 behaviours. A key experimental finding is that activation of DN1a neurons activates the downstream DH44 neurons, as assayed by GCaMP calcium imaging. This occurs only in the third instar and not in the second instar, in keeping with the development of sleep-wake and feeding separation. The authors also show that glucose metabolic genes are required in Dh44 neurons to develop sleep rhythm and that DH44 neurons respond differently in malnutrition or younger larvae.

Strengths:

Previous studies from the same lab have shown that sleep is required for LTM formation in the larvae, and that this requires DN1a and DH44 neurons. The current work builds upon this observation and addresses in more detail when and how this might develop. The authors can show that low quality food exposure and enhanced feeding during larval stage of *Drosophila* affects the formation of sleep rhythm and long-term memory. This suggests that the development of sleep and LTM are only possible under well fed and balanced nutrition in fly larvae. Non-sleep larvae were fed in low sugar conditions and indeed, the authors also find glucose metabolic genes to be required for a proper sleep rhythm. The paper presents precise genetic manipulations of individual classes of neurons in fly larvae followed by careful behavioural analysis. The authors also combine thermogenetic or peptide bath application experiments with direct calcium imaging of specific neurons.

Weaknesses:

The authors tried to induce sleep in younger L2 larvae with Gaboxadol feeding, however, the behavioral results suggest that they were not able to induce proper sleep behaviour as in normal L3 larvae.

Some of the genetic controls seem to be inconsistent. Given that the experiments were carried out in isogenized background, this is likely due to the high variability of some of the behaviours.

---

## [Author Response]

The following is the authors’ response to the original reviews.

**eLife assessment**
This study presents valuable findings as it shows that sleep rhythm formation and memory capabilities depend on a balanced and rich diet in fly larvae. The evidence supporting the claims of the authors is convincing with rigorous behavioral assays and state-of-the-art genetic manipulations. The work will be of interest to researchers working on sleep and memory.
**Public Reviews:**
Summary:This manuscript investigates how energetic demands affect the sleep-wake cycle in *Drosophila* larvae. L2 stage larvae do not show sleep rhythm and long-term memory (LTM), however, L3 larvae do. The authors manipulate food content to provide insufficient nutrition, which leads to more feeding, no LTM, and no sleep even in older larvae. Similarly, activation of NPF neurons suppresses sleep rhythm. Furthermore, they try to induce a sleep-like state using pharmacology or genetic manipulations in L2 larvae, which can mimic some of the L3 behaviours. A key experimental finding is that activation of DN1a neurons activate the downstream DH44 neurons, as assayed by GCaMP calcium imaging. This occurs only in third instar and not in second instar, in keeping with the development of sleep-wake and feeding separation. The authors also show that glucose metabolic genes are required in Dh44 neurons to develop sleep rhythm and that DH44 neurons respond differently in malnutrition or younger larvae.Strengths:Previous studies from the same lab have shown the sleep is required for LTM formation in the larvae, and that this requires DN1a and DH44 neurons. The current work builds upon this observation and addresses in more detail when and how this might develop. The authors can show that low quality food exposure and enhanced feeding during larval stage of *Drosophila* affects the formation of sleep rhythm and long-term memory. This suggests that the development of sleep and LTM are only possible under well fed and balanced nutrition in fly larvae. Non-sleep larvae were fed in low sugar conditions and indeed, the authors also find glucose metabolic genes to be required for a proper sleep rhythm. The paper presents precise genetic manipulations of individual classes of neurons in fly larvae followed by careful behavioural analysis. The authors also combine thermogenetic or peptide bath application experiments with direct calcium imaging of specific neurons.Weaknesses:The authors tried to induce sleep in younger L2 larvae, however the behavioral results suggest that they were not able to induce proper sleep behaviour as in normal L3 larvae. Thus, they cannot show that sleep during L2 stage would be sufficient to form LTM.

We agree that the experiments with Gaboxadol feeding in L2 did not perfectly mimic L3 sleep behaviors. However, genetic induction of sleep in L2 was effective in increasing sleep duration and depth similar to that observed in normal L3. As noted below in response to specific reviewer comments, because gaboxadol feeding is standard in the field for adult sleep induction, we prefer to still include this data in the manuscript for transparency. Moreover, the gaboxadol manipulation did cause a significant decrease in arousal threshold compared to control larvae. Together these approaches support the hypothesis that sleeping more/more deeply is not sufficient to promote LTM in L2.

The authors suggest that larval Dh44 neurons may integrate "information about the nutritional environment through the direct sensing of glucose levels to modulate sleep-wake rhythm development". They identify glucose metabolism genes (e.g., Glut1) in the downstream DH44 neurons as being required for the organization of the sleep-wake-feeding rhythm, and that CCHa signaling in DN1a signaling to the DH44 cells via the receptor. However, how this is connected is not well explained. Do the authors think that the nutrient sensing is only occurring in the DH44 neurons and not in DN1a or other neurons? Would not knocking down glucose metabolism in any neuron lead to a functional defect? What is the evidence that Dh44 neurons are specific sensors of nutritional state? For example, do the authors think that e.g. the overexpression of Glut1 in Dh44 neurons, a manipulation that can increase transport of glucose into cells, would rescue the effects of low-sugar food?

We thank the reviewer for these suggestions and have added the experiment proposed. We found that knockdown of Hex-C in DN1a neurons did not disrupt sleep-wake rhythms (Fig. S4G-I) suggesting that Dh44 neurons are specialized in requiring glucose metabolism to drive sleep-wake rhythms. We have also added further clarification in the text regarding the existing evidence that Dh44 neurons act has nutrient sensors.

Some of the genetic controls seem to be inconsistent suggesting some genetic background effects. In Figure 2B, npf-gal4 flies without the UAS show no significant circadian change in sleep duration, whereas UAS-TrpA flies do. The genetic control data in Figure 2D are also inconsistent. Npf-Gal4 seems to have some effect by itself without the UAS. The same is not seen with R76G11-Gal4. Suppl Fig 2: Naïve OCT and AM preference in L3 expressing various combinations of the transgenes show significant differences. npf-Gal4 alone seems to influence preference.

The sleep duration and bout number/length data are highly variable.

All experiments are performed in isogenized background so variability seen in genetic controls likely reflects stochastic nature of behavioral experiments. Indeed, adult sleep data also shows a great deal of variability within the same genetic background (PMID: 29228366). We agree it is an important point, and we attempt to minimize variability as much as possible with backcrossing of flies and tight control of environmental conditions.

**Recommendations for the authors:**

**Reviewer #1 (Recommendations For The Authors):**
Low sugar exposure and activation of NPF neurons might not induce the same behavioral changes. LS exposure does not enhance mouth hook movements, but overall food intake. NPF activation seems to enhance mouth hook movements, but the data for food intake is not shown. This information would be necessary to compare the two different manipulations.

We thank the reviewer for this suggestion. However, we elected not to perform food intake experiments with the NPF activation experiments. Since we are not directly comparing the low sugar and NPF manipulations to each other, we think that both experiments together support the conclusion that immature food acquisition strategies (whether food intake or feeding rate) limit LTM performance.

The authors write that the larval feeding assays run for 4 hours, can they explain why that long? Larvae should already have processed food within 4 hours, so that the measurement would not include all eaten food.

We clarified the rationale for doing 4 hour feeding assays in the results section. We did 4 hours on blue dyed food because initial experiments of 1 hour with control L3 at CT1-4 were difficult to interpret. The measurement does not include all of the eaten food in the 4 hours but does reflect more long-term changes in food intake.

Sleep induction with Gaboxadol seems to not really work - sleep duration, bout number and length are not enhanced, and arousal threshold is only slightly lower. Thus, the authors should not use this data as an example for inducing sleep behaviour.

We agree this approach did not have a large effect in larvae. However, because gaboxadol feeding is standard in the field for adult sleep induction, we prefer to still include this data in the manuscript for transparency. Moreover, the Gaboxadol manipulation did cause a mild (but significant) decrease in arousal threshold compared to control larvae. Gaboxadol feeding also caused a significant decrease in total body weight compared to control larvae indicating that even slightly deeper sleep could be detrimental to younger animals.

Activation of R76G11 with TrpA1 seems to work better for inducing sleep like behaviour. However, the authors describe that they permanently activated neurons. To induce a "normal" sleep pattern, the authors might try to only activate these neurons during the normal enhanced sleep time in L3 (CT13?) and not during the whole day. This might also allow larvae to eat during day time and gain more weight.

We apologize that this point was not clearer, but we did do acute activation of R76G11(+) neurons, as proposed by the reviewer. We have clarified the text to make this point.

It would be interesting to see how larvae fed with high sucrose and low protein diet would behave in this assay. Do the authors suggest that sugar is most important for the development of sleep behaviour or that it is a combination of sugar and protein that might be required?

We agree that feeding larvae a high sucrose and low protein diet would be interesting. However, we initially tried a low protein diet and observed significant developmental delays. Therefore, we are concerned that developmental defects on a high sucrose and low protein diet would confound behavioral results. Additionally, the Dh44 manipulations (glucose & GCN2 signaling) suggest that sugar is the most important for the development of sleep behaviors.

**Reviewer #3 (Recommendations For The Authors):**
The authors could discuss if the interaction between DN1a clock neurons and Dh44 neurons is mediated synaptic or by volume transmission following the extracellular release of the CCHa1 neuropeptide. They write that "the development of Dh44 neuronal competency to receive clock-driven cues" and that "DN1a clock neurons anatomically and functionally connect to Dh44" but a discussion about volume vs. synaptic signalling would be of interest.

We thank the reviewer for this suggestion. We revised the discussion to address this point.

line 223 " demonstrating that post-synaptic processes likely". It would be interesting to read a discussion on whether it is known if these are postsynaptic or peptide-mediated volume effects?

We added additional text to the discussion to address these points.

- The authors may want to include a schematic of the circuit and how its position in the general anatomy of the fly larva.

We thank the reviewer for this suggestion. We have added a model figure to Fig. S6.

"Dh44 neurons act through glucose metabolic genes" - consider rewording e.g. require glucose metabolic genes

We revised the text.

- line 45 "Early in development, young animals must obtain enough nutrients to ensure proper growth" - this is too general, many animals do not feed in early life-cycle stages (e.g. lecitotrophic development), consider rewording

We revised the text to be more specific.

- line 90 "however, L3 at CT1 consume more than L3 at CT12 (Figure S1A)" - typo CT13, also consider rewording to match the structure of the sentence before 'however, L3 consumed more at CT1 than at CT13'

We revised the text to fix this error.

- Line 111 "and loss of deep sleep" - how is deep sleep defined and measured in the larvae? It is not clear from the data or the text.

We revised the text to define deep sleep in the results section. We also have a description of how arousal threshold is calculated in the methods.

- In Figure 3B and G the individual data points are not shown

We did not show individual data points for those graphs because we are plotting the average percentage of 4 biological replicates.

Typo:Figure 1 legend "F, n = n=100-172 "

We revised the text to fix this typo.